# Rock glaciers across the United States predominantly accelerate coincident with rise in air temperatures

Andreas Kääb [1] & Julie Røste[1]

Despite their extensive global presence and the importance of variations in their speed as an essential climate variable, only about a dozen global time series document long-term changes in the velocity of rock glaciers – large tongue-shaped flows of frozen mountain debris. By analysing historical aerial photographs, we reconstruct here 16 new time series, a type of data that has not previously existed for the North American continent. We observe substantial accelerations, as much as 2–3 fold, in the surface displacement rates of rock glaciers across the mountains of the western contiguous United States over the past six to seven decades, most consistent with strongly increasing air temperatures in that region. Variations between individual time series suggest that different local and internal conditions of the frozen debris bodies modulate this overall climate response. Our observations indicate fundamental long-term environmental changes associated with frozen ground in the study region.

Around 11% of the Earth's land surface (15% for the Northern Hemisphere alone), or around 14 million km², is characterized by permafrost conditions[1–3], meaning that the ground is permanently frozen at depth. Non-polar mountain ranges account for about 30% of this permafrost area[4]. In these mountain regions, the total area of permafrost is roughly 20 times larger than the area covered by glaciers[5]. Permafrost, along with its ice content, serves as a critical hydrologic regulator[6–8] and impacts on various types of natural hazards[4,9]. The seasonally thawing active layers of mountain permafrost are becoming deeper, and permafrost temperatures are rising worldwide[4,10–12].

Rock glaciers[13,14] are often an indicator of past or present mountain permafrost. Active rock glaciers are suggested to be defined as expression of cumulative deformation by long-term creep of ice/debris mixtures under permafrost conditions but other definitions exist[14]. The results of such long-term slope deformation are distinctive, tongue-like landforms of frozen debris that move with surface rates in the order of centimetres to a few metres per year[15–17]. Warmer frozen debris typically moves faster than colder material, primarily because of the reduced viscosity of the warming ice core or ice-debris mixture, and due to the effects of infiltrating meltwater[18–20].

The variability of rock glacier velocities has recently been acknowledged by the World Meteorological Organization (WMO) as one product of the essential climate variable permafrost. Empirical data confirm the susceptibility of rock glacier velocities to climatic changes[16,18–21]. However, only a limited number of long-term time series documenting these changes – spanning several decades – currently exist. The majority of time series are located in the European Alps, with only a few in other mountains[16,22]. Despite the significant presence of rock glaciers in North America[23,24] and some velocity measurements available over shorter time intervals[25–29], no systematic long-term time series capturing the kinematic behaviour of rock glaciers have been established on this continent.

The mountainous regions of the contiguous U.S., encompassing roughly 15,000 km² of permafrost[1], host about 10,000 active rock glaciers[23], which cover a collective area of approximately 1000 km². These numbers, in comparison to approximately 5000 glaciers[5] occupying an area of 670 km² highlight the importance of rock glaciers and permafrost to the study region[8], and motivate the analysis of changes in rock glacier surface speeds.

Developing long-term time series of rock glacier speed (>30 years) is a challenging and laborious task. For a few rock glaciers

[1]Department of Geosciences, University of Oslo, Oslo, Norway. ✉e-mail: kaeaeb@geo.uio.no

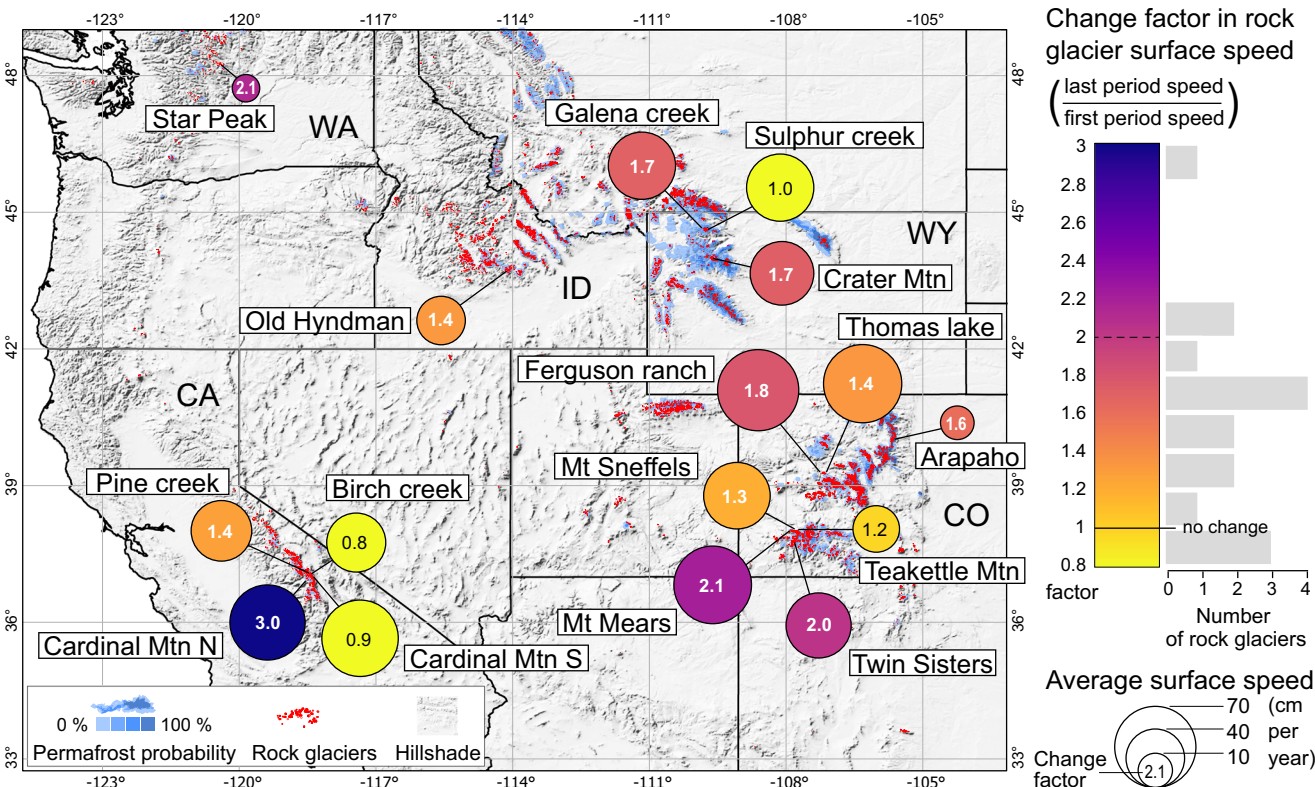

**Fig. 1 | Change in rock glacier surface speed between mid-end last century and recent decade over the western United States (U.S.).** Each circle indicates one rock glacier with its name defined in this study, the size of the circle its average horizontal speed, and its fill colour the change in speed (ranging from a slight deceleration in yellow to an acceleration factor of 3 in blue) between the first (mid to end of last century) and last measurement (recent one to two decades) of the speed time series compiled from repeat airphotos. Small red dots (due to high density in part appearing as red areas) are the positions of active rock glaciers from ref. 23 and blue areas are the modelled permafrost distribution from ref. 1. Background is a topographic hillshade derived from the Shuttle Radar Topography Mission elevation model (data courtesy of the U.S. Geological Survey), and the boundaries of states of the U.S. (black lines) with the state abbreviations given for states with rock glaciers studied here. Detailed rock glacier positions [lat°, lon°]: Star Peak [44.253, −120.417], Galena Creek [44.645, −109.791], Sulphur Creek [44.617, −109.756], Crater Mtn [44.023, −109.627], Old Hyndman [43.743, −114.106], Ferguson ranch [39.270, −107.194], Thomas lake [39.270, −107.155], Arapaho [40.020, −105.641], Mt Mears [38.018, −107.871], Mt Sneffels [38.010, −107.781], Teakettle Mtn [38.011, −107.768], Twin Sisters [37.762, −107.803], Pine creek [37.078, −118.450], Birch creek [37.065, −118.431], Cardinal Mtn N [37.011, −118.414], Cardinal Mtn S [37.008, −118.409]. Mtn: Mountain, Mt: Mount, N: North, S: South, WA: Washington, CA: California, ID: Idaho, WY: Wyoming, CO: Colorado.

worldwide, primarily in the European Alps, such series have been created from decade-long geodetic in-situ measurements[16,21]. However, these measurements cannot be retroactively extended, and accurately reconstructing rock glacier speed requires archived high-precision remote-sensing data. Satellite radar interferometry, currently used to measure seasonal rock glacier speeds, only yields data from the past around 20 years, and its application for decadal-scale time series is compromised by the disturbances from seasonal and inter-annual speed variations[17,29,30]. The only method to reconstruct >30-year long time series of rock glacier speed is by utilizing historical airphotos where they are available and accessible. The challenge lies in the fact that the substantial geo-positional inaccuracies that stem from camera types and calibration, and the resolution, radiometric quality, and distortions of these historic airphotos typically surpass the horizontal surface displacements of a few centimetres to metres associated with rock glacier movement[31,32].

Here, we digitally processed and analysed historical aerial photos to construct 60–70-year timelines of surface speeds for 16 rock glaciers located across the Rocky Mountains and the contiguous western United States. This type of data has not previously existed for the North American continent and more than doubles the global quantity of available long-term time series of rock glacier kinematics.

## Results

### Reconstructing rock glacier speeds from historical airphotos

A first objective of our study was to test and demonstrate how variations in rock glacier speed can be measured using the waste U.S. Geological Survey (USGS) airphoto holdings. We selected 16 rock glaciers that cover the latitudinal and longitudinal range of rock glaciers in the contiguous U.S. (Fig. 1), have sufficient airphoto coverage in the USGS archive, and exhibit clear signs of modern motion in Sentinel-1 radar interferograms[17]. (One of the 16 rock glaciers investigated, Hyndman, consists actually of two separate streams that are treated here as one system; Supplementary Information). USGS airphotos from approximately the 2000s onward are provided exclusively in ortho-rectified format (topographically corrected and projected to map coordinates). Prior to that time, only the original, non-rectified airphotos are available. We ortho-rectified the older airphotos from a challenging variety of camera and calibration types to compile a 60–70 year timeline of three to five orthophotos for each rock glacier studied. By exploring a large number of additional airphotos we found that more populated timelines are not feasible due to the limited existence of suitable airphotos with regard to spatial resolution, radiometric detail, and orthoprojection distortions ("Methods" section). We applied digital image matching between pairs of orthoimages to measure horizontal surface displacements between two consecutive

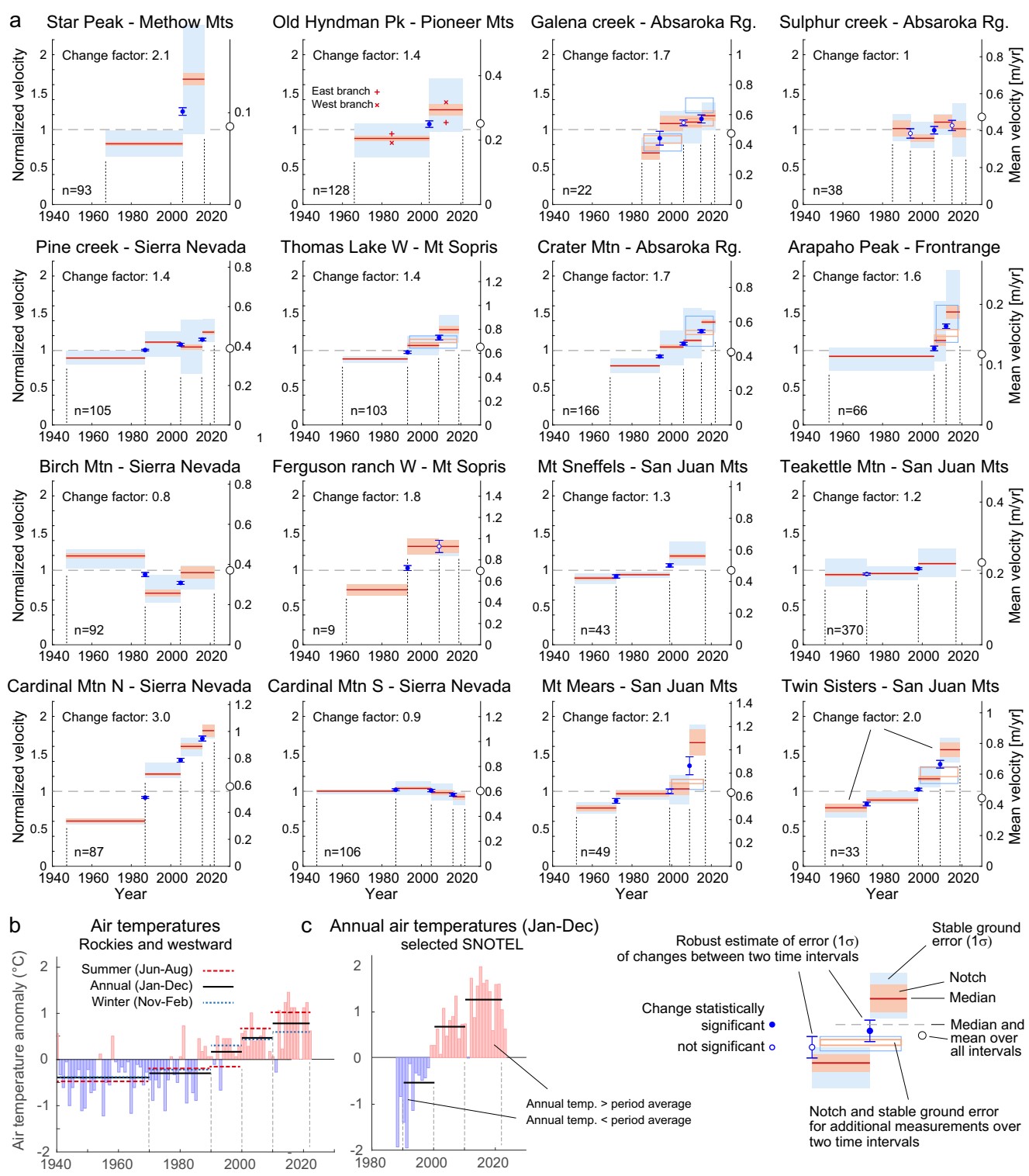

airphotos[31,33]. While this method was able to generate hundreds of successful displacement measurements when utilizing modern high-resolution high-precision orthoimages from USGS, the quantity and accuracy of successful displacement measurements significantly decreased for the old photos dating back to 1947 due to their low resolution and radiometric contrast ("Methods" section). To preclude any spatial bias, we retained only those measurement locations that yielded matches for all measurement periods. This still resulted in the creation of numerous individual time series of displacement rates on each rock glacier (*n* in Fig. 2; avg. 100, min. 9, max. 370). To prevent

result dilution from the large variability of absolute surface speeds over a rock glacier, each individual time series was normalized by its average speed before summarization for each rock glacier. We express the uncertainty of our results in three separate manners. Firstly, the notches between time-step median speeds (red in Fig. 2), next, the standard error of stable ground offsets after orthoimage co-registration (light blue), and finally a statistical significance test for changes between time steps (blue; "Methods" section).

Our study finds that and how decadal-scale time series of rock glacier speed can be reconstructed from the USGS archive, the – to our

**Fig. 2 | Detailed photogrammetric time series of horizontal rock glacier speed for the rock glaciers of Fig. 1 and regional air temperature data. a** Speed time series measured by automatically tracking surface features within repeated high-resolution airphotos. Horizontal red lines are the median speeds between two airphoto dates and light red areas are boxplot notches. Two medians differ at a >95% confidence interval if their notches do not overlap. Light blue areas indicate errors from feature tracking over stable ground around the rock glaciers (1 sigma). These light blue areas are vertically centred at the mean speed of the respective time interval, i.e., the less the light red and light blue areas are vertically symmetric the more is the displacement sample for the respective time interval affected by outliers. The blue error bars indicate the error of speed changes between measurement periods (1 sigma), and the blue circles on the error bars show the result of a Wilcoxon rank sum test of these changes (significant or not significant change). The change in speed between the first and last measurement period is given as change factor (last period / first period). '*n*' indicates the number of individual displacement time series from which the ensemble time series is compiled. The left axis indicates normalized speed (1 corresponds to the median speed), and the right axis the average of speeds over the n measurement locations (small circle on the axis indicates mean speed). Hyndmann Rock Glacier consists of two separate streams. The speeds of the individual parts are indicated by x (west) and + (east). **b** Mean measured air temperature anomalies for the climate region "Rockies and westwards" compiled by the National Oceanic and Atmospheric Administration (NOAA; "Methods" section) for winter, summer and full years, and decadal-scale means (horizontal lines) for time intervals similar to the airphoto intervals of panel a. NOAA temperature anomalies relative to 1940–2022 mean. **c** Mean measured air temperature anomalies for the Snow Telemetry Network (SNOTEL) high-elevation stations closest to study rock glaciers and decadal-scale means (horizontal lines). SNOTEL anomalies also relative to the 1940–2022 NOAA temperature mean. Mtn: Mountain, Mts: Mountains, Mt: Mount, N: North, S: South, W: West, Pk: Peak, Rg: Range.

best knowledge – globally largest freely and open available archive of airphotos.

## Variations in rock glacier speed

The average speeds on the investigated rock glaciers over the study period varied significantly, from 8 cm per year at the Star Peak rock glacier in the Methow Mountains (Washington state), to 70 cm per year at the Ferguson ranch rock glacier on Mount Sopris (Colorado). Maximum speeds at individual measuring points of up to 130 cm per year are found at some locations for Galena Creek rock glacier in the Absaroka Range (Wyoming). Our measurements are consistent with the few photogrammetric and geodetic measurements available on some of the rock glaciers studied ("Methods" section). Twelve out of the 16 rock glaciers showed a statistically significant acceleration over the measurement period, with several rock glaciers exceeding twice their initial speed between the first and final time intervals (Figs. 1 and 2a). One rock glacier (Cardinal Mountain North, California) even tripled its surface speed, from an average of 36 cm per year for 1947–1987 to 110 cm per year for 2016–2022. As per the extent that the availability of suitable orthoimages and thus the temporal density of the time series allows for such conclusions, several accelerating rock glaciers exhibit a consistent speed increase over time (e.g., Cardinal North, or Crater), with some others demonstrating a more hockey-stick like accelerated increase of speeds during the recent decades[22] (e.g., Mt. Mears, or Twin sisters; Fig. 2a). Conversely, the decelerating rock glaciers do so only slightly, close at or within the significance level of our measurements, with one even exhibiting acceleration during the more recent periods (Birch Mtn.).

## Discussion

Long-term alterations in air temperatures and ground-insulating snow are considered to be the most influential factors on permafrost temperatures and the deformation rate of frozen debris slopes, as warmer frozen debris typically moves faster than colder material[16,18–21]. Observed air temperatures over the contiguous U.S. climate region 'Rockies and westwards' have seen a strong increase since the 1990s. This warming is predominantly due to an increase in summer temperatures (Jun–Aug), around +0.4 °C per decade (Fig. 2b). High-elevation meteorological stations close to the rock glacier sites exhibit even more pronounced warming, up to about +0.8 °C and more per decade since the 1990s (Fig. 2c)[34]. Snow water equivalents at these elevations show less clear trends, larger spatio-temporal variability, and no strong overall temporal signal across the contiguous U.S. While there was an overall decline in snow cover since the 1980s (refs. 35–37), high-elevation snow stations to the east of the study region show rather increased snow water equivalent during recent years[35], and to the west rather decreased. Interpolated station data for the rock glaciers confirm the increasing air temperature trends while revealing a dip in annual and winter precipitation for southern rock glaciers and an uptick for northern counterparts in the last decade, with negligible changes in summer precipitation ("Methods" section and Supplementary Information).

The relationship between rock glacier motion and climatic change is multifaceted, influenced not just by the reduced viscosity of warming frozen materials but by a plethora of local variables. These include temperature and composition of material (e.g. debris grain size distribution and ice content); slope; thickness; vertical variation of material composition, in particular existence and depth of shear horizons; vertical profile of velocities, including velocity concentrations at layers in depth; water content; water pressure; lateral inflow/outflow of water; or the mass input to the rock glacier by, for instance, rock fall, snow avalanches and snow drift[18,20,21,25,38–41]. From the unknown but very likely variation of such local factors, it is thus well expected that not all of the rock glaciers investigated here respond in the same way.

Correlating the average speeds of the individual rock glaciers for the entire observation period in this study with the static rock glacier attributes from ref. 23 (Supplementary Information) unveils the highest coefficients for temperature-driven factors such as mean annual maximum ($R^2 = 0.46$) and mean annual temperatures ($R^2 = 0.38$). Nonetheless, correlations between the speed change factors and the static attributes are generally weak, the by far highest being with slope ($R^2 = 0.15$) and the others smaller than $R^2 = 0.06$. As source of uncertainty, the first measurement intervals, which serve as reference period for the change factors, are quite different among the rock glaciers studied (Table S1, Supplementary Information) and could therefore reflect different phases of climatic change.

Our time series of rock glacier speeds displays for most rock glaciers a quite high similarity ($R^2 > 0.5$) to mean annual or winter temperatures (Figs. 3 and S11–S13 in the Supplementary Information), but correspondence is less clear with summer temperatures. Correlations are weak for Mears, Sulphur and Birch, and clearly negative for Cardinal South. For precipitation (Figs. S14–S16, Supplementary Information), agreements are very variable and inconsistent, with some few good agreements (e.g., Galena), some negative correlations (e.g., Cardinal N and Twin Sisters), but mostly low correlations.

Cumulative evidence from laboratory experiments, numerical modelling, theoretical considerations[18–20,42,43], the global in-situ measurement series[16,21,22], and the predominant agreement between variations in rock glacier speed and air temperatures found in this study, all point towards a link between the observed rock glacier acceleration and the regional increase in air temperatures. Such acceleration can be driven by a decrease in the viscosity of the rock glacier material due to ground warming and the impact of meltwater from snow and ground ice[18–20,41].

The acceleration factors found here are in line with those theoretically predicted from a temperature-dependent ice deformation law and from a global set of rock glacier speeds[18]. Significant spatial variability in speed and speed changes among individual rock glaciers corroborates our understanding of the complex interplay of slope

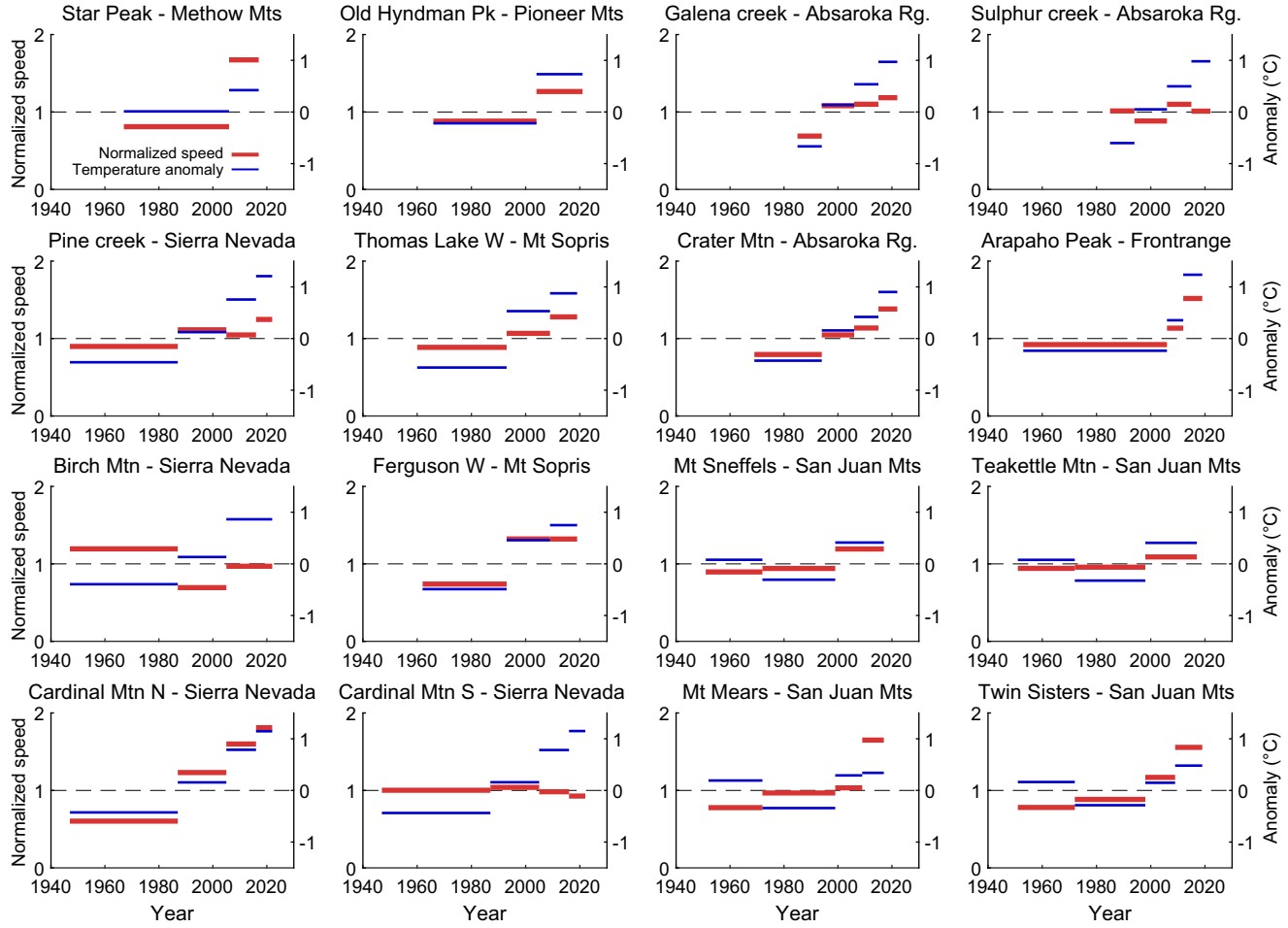

**Fig. 3 | Time series of normalized median rock glacier velocities (red) with anomalies of mean annual air temperatures (blue).** Speed time series are the same as in Fig. 2. The air temperatures are interpolated at the rock glacier locations and for the rock glacier elevations from the Parameter-elevation Regressions on Independent Slopes Model (PRISM) data set (PRISM Climate Group, Oregon State University, https://prism.oregonstate.edu, accessed 6 Jun 2024), and averaged for the same time intervals than the photogrammetric speed measurements. Mtn: Mountain, Mts: Mountains, Mt: Mount, N: North, S: South, W: West, Pk: Peak, Rg: Range.

gradient, thickness, and other factors listed above on individual rock glacier speeds and their response to environmental changes[18–21]. The detected rock glacier acceleration in the contiguous U.S. in general mirrors the trends observed on other continents, with some local rock glaciers exhibiting acceleration factors at the higher end of the global spectrum[16]. For most of our speed time series, comparable shapes of speed variations can also be found among the global set of time series[16,21,22]. Such similarities include hockey-stick-like accelerations, gradual accelerations over the entire measurement period, and variations in decadal speeds of different magnitudes without a strong overall accelerating trend. Cardinal South is the only clearly decelerating rock glacier out of the global long-term series from the present and previous studies, except of the, though, special case of a destabilized rock glacier in the western European Alps (Dirru)[21].

The acceleration of rock glaciers observed in this study, as for most of the other global long-term velocity time series, offers a pronounced contrast to the typical behaviour of mountain glaciers which usually slow under atmospheric warming, not least as a result of thinning of the ice bodies[44–47]. For instance, the absence of acceleration for Sulphur rock glacier and the recent deceleration in Galena rock glacier despite the regional warming might be attributed to recent slight reductions in rock glacier thickness[28]. Lastly, without data on thickness changes for the Cardinal South rock glacier, we can only speculate that its deceleration might be a consequence of ground ice loss.

By doubling the global number of long-term time series of the essential climate variable rock glacier surface speed, and constructing data that has not previously existed for the North American continent, we observe a substantial decadal-scale acceleration of rock glacier speed across the mountains of the western contiguous U.S. over the past 60–70 years. This behaviour is most consistent with strongly increased air temperatures in the region that can be expected to have decreased the viscosity of the frozen debris slopes through ground warming and meltwater infiltration. Despite the limited number of rock glaciers investigated, our results unequivocally suggest significant changes in the dynamics of rock glaciers across the contiguous U.S. These findings indicate an intensification of periglacial activity and a plethora of other possible environmental long-term changes associated with frozen ground in the study region that are even more challenging to quantify, such as alterations in ground temperatures, in fluxes of mountain debris and ground ice, in cryo-conditioned landscape development and mountain hazards, and in mountain hydrology and water resources.

## Methods
### Selection of sites
We selected 16 individual rock glaciers for our study following a combination of the following criteria. The study sites span the latitudinal and longitudinal range of rock glaciers in the contiguous U.S. in

order to let us investigate rock glacier behaviour under climatic and topographic gradients. Where feasible we selected clusters of several nearby rock glaciers within a given mountain range to also consider the variability in rock glacier behaviour under similar regional conditions. As a crucial criterion, the sites needed to have coverage by suitable airphotos in the U.S. Geological Survey (USGS) archive. This criterion encompasses the existence of low-altitude airphotos with spatial resolution and radiometric quality sufficient enough to enable the identification of detailed tracking features on the rock glaciers. This criterion affects in particular the first decades of our time series in the mid-20th century. In order to avoid laborious pre-processing steps for rock glaciers that then in the later analysis turn out to have a surface speed below detection thresholds of the old airphotos, we ordered and consulted satellite radar interferograms based on recent synthetic aperture radar (SAR) data from the Sentinel-1 mission. Several interferograms with temporal baselines of 12 and 24 days were chosen and ordered from the Alaska SAR Facility (ASF) Vertex service. For compilation of our speed time series, we selected rock glaciers that show clear signs of contemporary motion in the Sentinel-1 radar interferograms through motion-induced interferometric fringes or localized, and thus likely motion-induced, loss of interferometric coherence[17,31]. Following standard procedures for interferometry-based rock glacier classification, we relied in our assessment of contemporary rock glacier activity on a combination of interferograms from different times and with different temporal baselines[17,48]. Most interferograms used stem from Sentinel-1 data of 2022 and 2023. In case several rock glaciers appeared to be feasible sites in a region or mountain range following the above criteria, we preferred rock glaciers with earlier investigations available in order to aid the interpretation and validation of our results, and provide background for or motivate forthcoming studies.

## Pre-processing of airphotos

For times before roughly the 2000s only original non-rectified airphotos are available from USGS. These images have been scanned, or are scanned on demand, and stem from a challenging variety of camera and calibration types. We selected suitable images based on metadata (e.g., image scale) and visual inspection of quick-looks or full-resolution data (e.g., checking for lack of snow cover and sufficient radiometric quality and details). For each individual photo, we collected camera calibration protocols either directly attached to the scanned images, or from the USGS calibration protocol archive. We implemented these calibrations in the photogrammetric software PCI Geomatica OrthoEngine to solve for the inner orientation of each photo. To reconstruct the exterior orientation we chose approximately 30–50 ground control points[31] from the most recent orthorectified high-resolution image provided by USGS for horizontal control, and extract the elevations of these points from the Copernicus global 30m-resolution digital elevation model (DEM) for vertical control. The bundle adjustment for each photo or set of connected photos from the same flight gave typically residuals well below 1 metre for georeference. We then topographically corrected the photos within PCI Geomatica OrthoEngine and projected them to UTM coordinates with NAD datum using the image orientation parameters obtained and the Copernicus global DEM.

For about the 2000s on, airphotos are provided by USGS exclusively in ortho-rectified format. These images need no further pre-processing for our purposes. We selected suitable images from their spatial resolution, radiometric quality and detail, and with sufficiently long temporal baseline to enable statistically significant motion detection (i.e., about 5–10 years). We found that the most recent USGS orthoimages have excellent resolution, positional accuracy and precision, and radiometric quality which makes them very well-suited for our purpose. The older orthoimages are more affected by lower resolution and distortions (see below on error budget).

After building a 60–70 year long time series of three to five orthophotos for each rock glacier, we co-registered these orthophotos locally around the study rock glacier using image matching[33] (see next section on displacement measurements). For that purpose, we chose a set of 20–40 points outside of the rock glacier that we assessed to be lying on stable ground, based on our radar interferograms and interpretation of ground cover and geomorphology (e.g., exposed bedrock). The co-registration parameters were stored and applied to correct the raw displacement measurements.

## Displacement measurements

We measured displacements for individual targets on the moving rock glaciers between two orthoimages of different timestamps for a given rock glacier (typical intervals between orthoimages from several to about one decade) using automatic image correlation based on the normalized cross-correlation algorithm[33,49,50]. For the more recent images with high spatial resolution and radiometric detail such measurements would be possible for dense grids of points, but such procedure fails for most locations for the older images that depict much fewer visual targets such as individual boulders, or furrows and ridges, that can be detected and tracked between images. For grid-based matching[33], this problem results in a much smaller number of successful matches than mismatches, which cannot be completely removed through filtering and thus result in substantial contamination of the fields of measured displacement vectors by outliers. We therefore manually selected a large number of irregularly spaced locations on the rock glaciers that contain features that can be identified in all orthoimages of a time series and automatically measure displacements there, i.e., we visually prescribe matching positions. These individual locations are in most cases well distributed over the entire rock glacier (Supplementary Information). The image matching software we were using for this purpose[33] allows for interactive selection of matching points, and immediate point-wise automatic image matching and display of displacement vectors. Gross outliers can in that way directly be identified visually and interactively removed. For the image matching we used template sizes of between 20 and 25 metres and search areas were determined iteratively to capture the full range of actual displacements. We measured displacements between all subsequent orthoimages of a series. For some rock glaciers where we found particularly uncertain results, we measured displacements also for additional orthoimage pairs spanning two instead of one-time interval.

## Speed time series

Once all measurements were conducted for all orthoimages over a rock glacier we built the velocity time series and extracted the vector magnitudes to obtain speeds. Based on the displacement locations of the earliest time interval, which typically contained the least successful measurements due to the lowest image quality, we selected for each initial location the measurement points in all other subsequent displacement fields closest to the initial one, if existing within a radius of 40 m. In other words, the starting points of measured displacement vectors that are closest to a starting point of displacements in the first image pair and in addition within a 40 m radius to this initial starting point are considered to represent the same location on the rock glacier. All other measurements in the image pairs after the first one are disregarded. Choosing this procedure over using a fixed set of measurement locations throughout the time series allows for the selection of optimal measurement locations that contain features of high contrast for each image pair individually. Each initial measurement location within the earliest image pair that has close-by measurements for all other time intervals defines then one-speed time series. To avoid that the large spread of displacement magnitudes over a rock glacier dilutes our results, we normalize each individual time series by its mean speed.

## Error budget

The largest component of the error budget of our time series are typically the unresolved orthoimage distortions[31]. Such distortions can stem from distortions within the original airphotos (e.g., lens and camera distortions, distortions of the photographic film, or scanning distortions), from distortions of the photogrammetric model, and from distortions within the orthoprojection process. The latter errors come typically from inaccurate DEMs used for orthoprojection. Whereas these distortions can be evaluated for stable ground as the expected ground motion there is zero, they cannot be separated from the actual ground motion on the rock glaciers[31,51] and therefore not be corrected for. The other error budget component of concern are matching errors, i.e. the precision with which a matched displacement corresponds to a (typically unknown) real ground displacement[31,52]. Also this error can be evaluated from stable ground offsets, but on the moving rock glacier, it can only be evaluated from validation data. Both error components affect measurements on the rock glaciers mainly in a random way as systematic offsets are reduced by the image co-registration step. Through choosing ground control points (GCPs), which are necessary to solve the photogrammetric bundle adjustments, from the latest USGS orthoimages and their elevation from the Copernicus global DEM, and through co-registering the orthoimage pairs, effects from absolute geo-reference errors can only have very minor second-order impact on our time series, and only in the presence of strong local spatial gradients in rock glacier speed. A threshold on spatial speed gradients could be employed to mask out measurement locations with strong local gradients. Such procedure was not applied as we avoided to measure displacements close to rock glacier margins and did not encounter break lines in the displacement fields on the rock glaciers investigated, as for instance confirmed by flickering between images of different times.

As a result of the above error processes, we express the uncertainty of our results in several independent ways. First, the speeds for each time interval and each rock glacier are displayed as median, a robust measure that reduces the effect of outliers, including outliers from local distortions and matching errors. The generally small deviations between these medians and the time interval means confirm the absence of large outliers. Based on the medians and the spread of normalized speeds for each series and time interval, notches are computed. Notches[53] of subsequent time intervals that do not overlap indicate that their medians are statistically different at a significance level of >95 %. Second, we compute and show the estimated standard error from stable ground offsets. For that purpose, we compute the normalized median absolute deviation of stable ground offsets for each time interval and rock glacier, and from this absolute deviation, we derive a robust estimate of the standard deviation of stable ground offsets[54]. To then estimate from this standard deviation (i.e. dispersion of the offsets) the standard error for stable ground offsets (i.e., uncertainty of the offset mean) we conservatively assume our 20–40 stable ground measurements to represent only 4 independent measurements (i.e., divide the standard deviation by $\sqrt{4}$). This reduction accounts for the fact that neighbouring stable ground offsets could be affected by the same low-frequency distortions and could thus be partially correlated. Third, we compute the standard error of speed changes between time intervals for a specific rock glacier and statistically test whether the speed changes are statistically significant. For the latter test, we use the Wilcoxon rank sum test (equivalent to the Mann–Whitney $U$-test), roughly speaking a robust version of the student test.

## Climate data

At the decadal time scales considered in this study, air temperature data from a range of sources show similar temporal variations and trends. We collect and assess annual and seasonal air temperature time series for all our mountain ranges studied using the NOAA compilation of measured air temperatures (https://www.ncei.noaa.gov/access/monitoring/climate-at-a-glance/regional/time-series) and measured temperatures at SNOTEL stations as close as possible to the rock glaciers studied (https://nwcc-apps.sc.egov.usda.gov/imap). We also visually check climate re-analyses such as ERA5 as prepared on the University of Maine Climate Reanalyzer (https://climatereanalyzer.org/research_tools/monthly_tseries). The second climate variable that is considered to be able to significantly impact rock glacier speed is snow cover, through insulating the ground in winter and thus potentially reducing seasonal ground cooling. Snow water equivalents at high elevations over our study region show no clear trends and large spatio-temporal variability. We assess the snow water equivalent measured at SNOTEL stations as close as possible to the study rock glaciers (https://nwcc-apps.sc.egov.usda.gov/imap) (Supplementary Information). We also extracted the winter (Oct–Mar) snowfall water equivalents and winter total precipitation from the ERA5 re-analysis[55] in two ways. First, we summarize both variables for all elevations above 2000 m (geopotential height) over the western U.S. Second, we extract both variables specifically at the mountain ranges studied.

Measured air temperatures and precipitation, compiled and interpolated by the PRISM group (PRISM Climate Group, Oregon State University, https://prism.oregonstate.edu) were extracted for the locations and elevations of the individual rock glaciers (Supplementary Information).

## Comparison to other velocity measurements

There are only few geodetic in-situ measurements available to compare our results to. In-situ total station measurements on Arapaho rock glacier between 1960, 1985 and 2002 for a transect in the middle of the rock glacier (points 1–4 in ref. 27) reveal average speeds of 14.5 cm/yr, our speeds over 1953–1999 for the same location give 13.6 cm/yr. Both measurements agree by far within their uncertainties. The results by ref. 27 also show that no significant decadal-scale speed variation seems to have happened within our 1953–1999 period. Photogrammetric displacement measurements on Arapaho rock glacier[32] over 1978, 1990 and 1999 (e.g., 12.8 cm/yr for the above transect) also agree well with our measurements and indicate little change over 1978–1999.

Comparing in-situ geodetic measurements of boulder displacements on Galena Creek rock glacier[26,28] over 1997–2015 to displacements at nearby locations from our photogrammetric measurements of 1994–2015 gives a mean absolute deviation of 3 cm/yr, which is well within the notches and errors of around 10 cm/yr that we give for this rock glacier and period. The in-situ speed variations between 1997, 2015 and 2022, and photogrammetric measurements 2020–2022, also from ref. 28, agree with our photogrammetric measurements within their error bounds, all consistently indicating a slight increase in speed during the recent years.

Satellite radar-interferometric measurements over 2007–2008 by ref. 29 are difficult to compare to our measurements due to the different temporal and spatial scales, and the different measurement locations involved. Mean radar-interferometric speeds for late summer 2007 for Birch Mountain rock glacier are 45 cm/yr (from ref. 29; our measurements 2005–2022: 35 cm/yr), for Pine Creek 57 cm/yr (our measurements 2005–2016: 41 cm/yr), for Cardinal North 61 cm/yr (our measurements 2005–2016: 94 cm/yr), and Cardinal South 67 cm/yr (our measurements 2005–2016: 61 cm/yr).

## Data availability

USGS orthoimages or original airphotos and their camera calibrations are openly available from https://earthexplorer.usgs.gov. Derived displacements from the current study are openly available from the Zenodo repository under accession code https://doi.org/10.5281/zenodo.13254373 (ref. 56). Climate data used are openly available from NOAA and the Copernicus Climate Data Store[55] (https://cds.

climate.copernicus.eu/), SNOTEL data from NWCC, and the PRISM data from Oregon University (https://prism.oregonstate.edu/). The Copernicus global DEM is openly available from the European Union PANDA service (https://panda.copernicus.eu/).

## Code availability

PCI Geomatica OrthoEngine is a commercial software. The image matching software used (CIAS) is freely available from https://mn.uio.no/icemass.

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

## Acknowledgements
This study would not have been possible without the open data policy for USGS airphotos and orthoimages, and the agency's invaluable collection and processing of these data. We would also like to thank other providers of open data including SNOTEL, PRISM, other climate data and Copernicus DEM. This study was supported by the European Space Agency projects Permafrost_cci, Glacier_cci, and EarthExplorer10 Harmony (4000123681/18/I-NB, 4000127593/19/I-NB, 4000135083/21/NL/FF/ab) and the University of Oslo dScience Centre.

## Author contributions
A.K. designed the study, performed the photogrammetric measurements, analysed the data, and wrote the text. J.R. prepared climate data and edited the text.

## Competing interests
The authors declare no competing interests.
