## [Peer Review File · Nature Communications]

Rock glaciers across the United States predominantly accelerate coincident with rise in air temperaturesREVIEWER COMMENTS

Reviewer #1 (Remarks to the Author):

I enjoyed reading this interesting manuscript. The complex behavior of rock glaciers in response to climate change can only be decoded with more long-term series on rock glacier creep. With this timely study on rock glacier speed and its variation for the contiguous United States, the global picture of rock glacier kinematics and its temporal changes is enhanced significantly. This is urgently needed to properly quantify and assess the new product “rock glacier velocity” of the Essential Climate Variable “Permafrost”. Many thanks for this important study!

The manuscript is very well written, structured and illustrated.

The motivation for this original study is clearly elaborated and credit is given to important studies in the context of rock glacier behavior in relation to increasing temperatures. New series on rock glacier speed are provided for 16 landforms in different regions of the contiguous United States, a region that was lacking long-term information so far. The conclusion derived from this study is in line with findings from other regions, supporting the observation of general rock glacier acceleration. The fact that some landforms (4 out of 16) show a different trend is also in line with other studies and underlines the need for more detailed analyses of influencing factors on rock glacier creep. As the data series are compiled by a common methodology, they are easily comparable to series from other regions (e.g. from the European Alps). For sure, these noteworthy results will be included in future global assessments.

The methodology of processing aerial imagery and deriving time series of rock glacier speed is described in a very detailed and concise way, allowing reproducibility. A proper error assessment is provided. The additional use of InSAR data for the site selection as well as comparisons with in-situ geodetic measurements from some site shows the full range and potential of information on rock glacier movements and the value of complementary data. The outcomes are nicely summarized in two figures showing the spatial and temporal patterns of rock glacier speeds.

Few individual edits are listed below.

I 52: “...geodetic in-situ measurements” add reference to the most recent study by Kellerer

et al. 2024: Acceleration and interannual variability of creep rates in mountain permafrost landforms (rock glacier velocities) in the European Alps in 1995–2022
(<https://doi.org/10.1088/1748-9326/ad25a4>)

l 70: “insufficient co-registration accuracy”, please provide a reference

l 174: delete “at”

Figure 1: The signature for “Rock glaciers” is hardly readable in the legend, please enlarge. If possible, be more explicit regarding the decades you compare “mid-end last century” vs “recent decade” (= 2010-2020?). How large is the deviation of the time periods you compare?

Figure 2: Would be nice to be able to read the decades more clearly; please add thin lines for each decade (like in Figure 2b and 2c)

Supplementary material:

Figure S1 and S2: provide the years of the “entire measurement period” (possibly in the captions, as it seems to be the same for all the stations) and add information on the data source (NOAA, NRCS)

Figure S3 and S4: it is not fully clear whether you show snow-water equivalent or winter total precipitation; add description and unit of the axes. Provide data source.

Reviewer #2 (Remarks to the Author):

The work suggests temporal and spatial displacement signals from a sample of rock glaciers. The study found significant changes in velocities for some of the rock glaciers investigated. In the western part of the study area rock glaciers both accelerated and decelerated. In the eastern and northern parts mostly, acceleration was observed. Therefore, in my opinion, the work does not fully support what is claimed in the title “Creep of rock glaciers accelerates across the United States”. My advice is that the title should be adjusted accordingly. The conclusion (line 129-131) is more in line with the variability in the data “suggesting significant changes” instead of just acceleration.

I also miss data and discussion regarding variation in local snow cover and precipitation for

each rock glacier. It would also be useful if the authors could present mean measured air temperature anomalies closer to the rock glaciers dating further back in time than the SNOTEL-data presented in Fig2b. It would also be valuable if the authors could compare the observed displacement patterns to other available long-term studies.

All in all, the authors present convincing results, discussions and conclusions based on substantial data and sound use of methods. The time span covered and details both spatially and temporally are impressive and, in my experience, unprecedented and will have a large significance to the field and similar fields.

See my detailed suggestions and comments in the attached annotated manuscript.

Comments/suggestions for Supplementary Information:

Fig S2: Consider naming the RGs that are close to the stations.

Fig S3: Consider adding unit to y-axis.

Fig S4: Consider adding unit to y-axis.

Best regards,

Harald Øverli Eriksen

Reviewer #3 (Remarks to the Author):

There are several problems with this paper and it should not be published in its present form.

1 The submission is to Nature Communications. However, the title does not convey what is necessary to communicate. The text is mainly about the techniques involved and data extraction. While I have no complaints about the technique, it is novel and useful, indeed, I would like to know more about it and results that can be seen. But this journal is not the place to present this. The technique and basic results should be published elsewhere.

2. This title is not only misleading in intent but also in fact, some 'self-evident' others more nuanced. It is not 'creep' that is necessarily being measured but 'speed' or 'velocity' and these words are used in the text. Creep is associated here with rock glaciers, as in text and as stated but also with glacier ice deformation. Glaciers can slide to give a surface velocity.

3. The linkage of rock glacier to 'permafrost creep' should be abandoned. This has long been used as the only mechanism for rock glacier flow, usually following citation of the book Rock Glaciers by D. Barsch (here reference number 14). Barsch denies the existence of the 'glacier ice cored rock glacier'. In particular the work of Potter at Galena Creek RG where glacier ice has been shown to be the core of the rock glacier at Galena Creek and at Sulphur Creek RG the recent work by Meng et al (cited in the paper as ref 27) not only gives velocity data (was this compared to that obtained in this paper?). In other words, the authors use rock glacier always as being indicative of permafrost rock glaciers but omit references to evidence from 'disproving cases'. This has been a situation for far too long in the literature. Here the glacier ice formation is blatantly denied but still uses RG that do have glacier cores in the 'analysis'.

4. Any body with ice (glacier or cryogenic) can 'flow' or 'creep' but the creep rate (velocity) depends upon more than 'temperature'. Thickness of the ice mass and surface slope provide the main stress fields. So, a glacier might give the same results as supposed for a 'permafrost rock glacier'. But the problem really is that the data analyses are far too simplistic, other site specific conditions should be analysed. Further, if the rock glaciers are buried glaciers then sliding, rather than flowing, might be the main driver of the differences.

6. But what are these differences? Why does Cardinal Mtn N have the largest surface speed change compared with Cardinal Mtn S have one of the lowest? This alone might suggest that there is more in the variability than 'temperature'. Indeed, several meteorological parameters should be used and investigated. This leads to

7. The data analysis is overly simplistic. In fact, the paper does not comply with any aspects of FAIR data principles. Ok, have a graph but what are the data from which the results are plotted? Use of median might be more 'robust' but there are other descriptive statistics that could be used for the analysis.

8. Also with respect to FAIR, the locations of rock glaciers are given only by toponyms. A proper geolocation is required. Using a decimal latitude-longitude [dLL] identifier for rock glacier locations allows the reader to locate the feature in, for example Google Earth. This might help in interpretation, what might the differences be between Cardinal Mountain N/S. From the peak at [36.9996,-118.4148], there are several contenders for 'north' but I cannot see what is meant by 'south'. This is not presenting data that are findable, accessible, interoperable or re-usable. This is required in any transparent data analysis. No attempt to be transparent is made in this paper.

But fundamentally, it is the continued scientific lack exactitude of the data presented. Support for the contention suggested by the title is not upheld.

Rock glaciers across the United States predominantly accelerate coincident with rise in air temperatures

Andreas Kääh, Julie Røste

Response to referees and Revisions made
--

General response

We would like to thank the three referees for their thoughtful and constructive reviews that certainly helped to improve the paper. We implemented all recommended changes and in particular revised the main text and figures, expanded the analyses and discussion, and added substantially more information to the Supplement.

In general response to ref#3 we want to briefly comment here on the connection between permafrost and rock glaciers. This is a long-standing dispute in the scientific community that we are not involved in and try to avoid. Our interest lies in the kinematics of rock glaciers and its response to climatic changes. We believe our results are important, irrespective of the view the reader has on the connection between permafrost and rock glaciers. *Permafrost* is solely a thermal definition, *glaciers* are defined by form, material (ice), origin (snow) and movement. Whether rock glaciers are connected to permafrost or glacier ice does thus not have to be a contradiction or topic for dispute. When writing about rock glaciers in permafrost environments we follow a large body of literature and, for instance, the World Meteorological Organization (WMO) that we cannot just ignore. This is confirmed by ref#1 and ref#2 who do not criticize our terminology. On the other hand, we also want to acknowledge the existing literature that points to rock glacier like landforms that do not necessarily reflect permafrost conditions, or transitional landforms. We tried now to reach a more balanced terminology and discussion of processes (see also response to ref#3) that we hope can satisfy to some extent both view points on the connection between permafrost and rock glaciers that exist in the literature. We believe it should not be demanded from an article about (undoubted) rock glacier kinematics to solve a many decades old scientific dispute that is not the focus of the study.

See a more detailed discussion of the topic in Berthling (2011)(see reference list of the paper).

Editor/referee comments are in *italic*, and our response in normal font.

An annotated version of our revised manuscript with track changes is attached.

Response to reviewer #1 page 2

Response to reviewer #2 page 3

Response to reviewer #3 page 7

Response to Reviewer #1

I enjoyed reading this interesting manuscript. The complex behavior of rock glaciers in response to climate change can only be decoded with more long-term series on rock glacier creep. With this timely study on rock glacier speed and its variation for the contiguous United States, the global picture of rock glacier kinematics and its temporal changes is enhanced significantly. This is urgently needed to properly quantify and assess the new product “rock glacier velocity” of the Essential Climate Variable “Permafrost”. Many thanks for this important study!

The manuscript is very well written, structured and illustrated. The motivation for this original study is clearly elaborated and credit is given to important studies in the context of rock glacier behavior in relation to increasing temperatures. New series on rock glacier speed are provided for 16 landforms in different regions of the contiguous United States, a region that was lacking long-term information so far. The conclusion derived from this study is in line with findings from other regions, supporting the observation of general rock glacier acceleration. The fact that some landforms (4 out of 16) show a different trend is also in line with other studies and underlines the need for more detailed analyses of influencing factors on rock glacier creep. As the data series are compiled by a common methodology, they are easily comparable to series from other regions (e.g. from the European Alps). For sure, these noteworthy results will be included in future global assessments.

The methodology of processing aerial imagery and deriving time series of rock glacier speed is described in a very detailed and concise way, allowing reproducibility. A proper error assessment is provided. The additional use of InSAR data for the site selection as well as comparisons with in-situ geodetic measurements from some site shows the full range and potential of information on rock glacier movements and the value of complementary data. The outcomes are nicely summarized in two figures showing the spatial and temporal patterns of rock glacier speeds.

We would like to thank the reviewer for their positive and encouraging feedback. All their detailed comments have been incorporated as described below.

Few individual edits are listed below.

l 52: “...geodetic in-situ measurements” add reference to the most recent study by Kellerer et al. 2024: Acceleration and interannual variability of creep rates in mountain permafrost landforms (rock glacier velocities) in the European Alps in 1995–2022 (<https://doi.org/10.1088/1748-9326/ad25a4>)

Now cited at several locations. Thanks for pointing to this important study that came out in parallel to the first submission of the present study.

l 70: “insufficient co-registration accuracy”, please provide a reference

We clarified now that this statement is our own finding from exploring a large amount of airphotos and we referred now to more description in the Methods section.

l 174: delete “at”

Done

Figure 1: The signature for “Rock glaciers” is hardly readable in the legend, please enlarge.

Done

If possible, be more explicit regarding the decades you compare “mid-end last century” vs “recent decade” (= 2010-2020?). How large is the deviation of the time periods you compare?

Good point. The spread of time periods is actually quite large. We rewrote to “mid to end of last century” and “recent one to two decades”. The first periods range, for instance, between 1947-1987 and 1985-1994, the last periods between 1998-2017 and 2016-2022. We now list the periods for each individual rock glacier in the Supplement, and now list this spread and its consequences as source for uncertainty in our Discussion section.

Figure 2: Would be nice to be able to read the decades more clearly; please add thin lines for each decade (like in Figure 2b and 2c)

Vertical lines included.

Supplementary material:

Figure S1 and S2: provide the years of the “entire measurement period” (possibly in the captions, as it seems to be the same for all the stations) and add information on the data source (NOAA, NRCS)

Measurement periods and data source now included in the captions of Figures S1 and S2. Also, we specified now which rock glaciers are closest to the SNOTEL stations shown.

Figure S3 and S4: it is not fully clear whether you show snow-water equivalent or winter total precipitation; add description and unit of the axes. Provide data source.

Both fixed now.

Response to Reviewer #2 (H.Ø. Eriksen)

The work suggests temporal and spatial displacement signals from a sample of rock glaciers. The study found significant changes in velocities for some of the rock glaciers investigated. In the western part of the study area rock glaciers both accelerated and decelerated. In the eastern and northern parts mostly, acceleration was observed. Therefore, in my opinion, the work does not fully support what is claimed in the title “Creep of rock glaciers accelerates across the United States”. My advice is that the title should be adjusted accordingly. The

conclusion (line 129-131) is more in line with the variability in the data “suggesting significant changes” instead of just acceleration.

We modified the paper title to indicate that most but not all rock glaciers accelerated.

I also miss data and discussion regarding variation in local snow cover and precipitation for each rock glacier. It would also be useful if the authors could present mean measured air temperature anomalies closer to the rock glaciers dating further back in time than the SNOTEL-data presented in Fig2b.

In addition to the snow water equivalent (SWE) and temperature data of SNOTEL stations as close as possible to the rock glaciers in the Supplement, we include already average regional temperatures from NOAA (1940-2022) (Fig 2b). Now, we now include in the Supplement annual/summer/winter time series of temperatures and precipitation (1940-2022) interpolated for the individual rock glaciers by the PRISM Climate Group. We add now in the main text a figure (Fig 3) that shows speed changes and air temperature changes together. The PRISM data have also been used for the rock glacier inventory by Johnson et al. (2021) and are to our best knowledge indeed the best-suited data for our purpose as the PRISM group compiled a very large number of station data from a very large set of sources over the US.

It would also be valuable if the authors could compare the observed displacement patterns to other available long-term studies.

Such discussion is now extended on in the Discussion section.

All in all, the authors present convincing results, discussions and conclusions based on substantial data and sound use of methods. The time span covered and details both spatially and temporally are impressive and, in my experience, unprecedented and will have a large significance to the field and similar fields.

Thanks for this encouraging feedback!

See my detailed suggestions and comments in the attached annotated manuscript.

In the following we list the comments in the pdf (with the line number referring to the original pdf) and give our response:

Line 62: It would be of great value if you for each RG could give more information/context in the Supplementary Information. This could including maps/orthophotos showing the main features and spatial (and if possible temporal) distribution of the displacement. With this context it would be easier for the reader to relate/interpret the data in the other figures.

We added now in the Supplement one sheet for each rock glacier with airphoto and other information/context.

L86: From Fig. 1 maximum speed for Galena creek RG looks lower than 130 cm, see Fig 2a.

The 130 cm/yr refer to a *few individual* measuring points, whereas the data in Figs 1 - 3 refer to average and median speeds of *all* measuring points for an individual rock glacier. This is now clarified.

L94: Does this apply to Arapaho Peak RG also (see Fig 2a)?

As the first measurement period for Arapaho is exceptionally long due to a lack of sufficient airphotos (1953-2006) we prefer to not interpret the shape of the speed series and leave the text as is.

L97: It is hard to interpret significance levels in Fig 2a due to small graphs and symbols.

The pdf resolution is now enhanced so that these symbols should now be readable. We enlarged fonts where possible.

L111: Consider to add references here.

Done

L115: Have you considered the impact increased precipitation as rain would have on the acceleration?

In the literature air temperature and snow cover are suggested to be the most important potential factors for long-term changes of rock glacier speed. We now show also precipitation time series for the individual rock glaciers in the Supplement and include precipitation effects in the Discussion section.

L127: Consider to introduce variations in precipitation as rain for the RGs in the study. Modeled data or data from meteorological stations nearby could give a more detailed understanding of the observed displacement patterns. If you reduce the number of RGs plotted in Fig2a as suggested (see details in my comments in the figure label) there will be room for plotting precipitation-trends as a line in the plots by adding another a secondary axis (preferably with a different color for both line and axis to increase readability).

Rain data are now included, see above. We experimented with showing only a selection of rock glaciers in the main text but we really prefer to show all. We believe the strength of the study is both the number of rock glaciers studied in one and the variation of responses seen. A selection would not draw this picture very well in our view. We added now a figure with simplified speed changes (no error bars etc.) and air temperature as the latter clearly shows the largest explanatory power. Precipitation trends are now shown in the Supplement and correlations have been investigated.

L141: Consider to add change in speed in each circle as suggested for Cardinal Mtn N (see red arrow). It can be hard, especially for people with reduced color vision (~8% of males!), to interpret only using the color legend.

We tested the figure for all kinds of reduced colour vision, and it should be ok, but find the annotation a good idea that we happily implement. We included the change factor. We also simplified Fig. 2 to contain only change factors, not anymore percentage change, as they are redundant.

L150: I think the plots in Fig 2a are too small to be interpreted. Details are blurred. Especially for additional measurement over two time intervals. Suggestions to consider: - Reduce the number of RGs plotted in Fig2a to maybe 4, leaving the rest for the Supplementary Information. Or show larger versions of the plots in the Supplementary Information. If the plots are increased there will be room for plotting precipitation-trends as suggested.

See above response to L127.

- To increase readability and better link the RGs between Fig1 and Fig 2a, in the heading of each RG in Fig2a, name the RG than the mountain range. Ex: Thomas Lake W - Mt Sopris.

Done

L153: Consider to add some info what the notches show, ex: "boxplot notches (95% confident interval of the median)"

Now explained in the figure caption.

L173: Consider to define and reference CONUS.

Changed to contiguous U.S.

L249: Could it be that you have missed some high-speed parts an RG that have a displacement above 40 m between acquisition of the orthophotos?

Sorry, a misunderstanding. These 40 m do not refer to a maximum displacement threshold but to a radius within which exact measurement locations (which differ from image pair to image pair to follow locations of suitable matching features) are considered to represent the same measurement location. We clarified.

L269: Is it possible to say something about the spatial speed accuracy, or quantify, a spatial gradient speed threshold?

If a measurement location is wrong in one image pair by some meters the measured displacement would still be very similar to the one some meters away. The stress transfer by ground ice makes the velocity fields on rock glaciers typically quite smooth. Such geolocation errors could only have an impact at the very margins of the rock glacier where a measurement location could jump on/off the moving rock glacier over a very short distance, or in the case of strong break lines in the flow field. We avoided to measure displacements on the rock glacier margins (see new rock glacier sheets in the Supplement) and became not aware of strong breaks in the flow fields of the rock glaciers investigated. These effects would be random. But, yes, a gradient threshold within the displacements of one time interval could help to exclude measurement zones prone to these effects. We added a respective sentence in the text.

Comments/suggestions for Supplementary Information:

Fig S2: Consider naming the RGs that are close to the stations.

Done in the caption of Fig S1

Fig S3: Consider adding unit to y-axis.

Fig S4: Consider adding unit to y-axis.

Done

Response to Reviewer #3

There are several problems with this paper and it should not be published in its present form.

We significantly modified our text along the referee's comments and would be happy to learn if there are formulations remaining that could lead to confusion and distract from what we believe are the most important results from our study: the kinematic time series. See in particular the general response at the top of this response letter.

1 The submission is to Nature Communications. However, the title does not convey what is necessary to communicate. The text is mainly about the techniques involved and data extraction. While I have no complaints about the technique, it is novel and useful, indeed, I would like to know more about it and results that can be seen. But this journal is not the place to present this. The technique and basic results should be published elsewhere.

We have now modified the title and the paper significantly, including an extension of the discussion. The technique used here builds on Kääh et al. (2021; see reference list of the paper) which can be consulted for other photogrammetric details. We hope the technique presented in the current study inspires similar studies elsewhere and gives useful instructions for that. In agreement with referees #1 and #2, we believe the most novel results from our study are the time series of rock glacier speed themselves. These data double the amount of such time series measured today, reach particularly far back in time, and cover a continent without such data available so far. We believe these findings are important enough for a number of fields, ranging for instance from the mechanics of frozen materials to climate change impact, to warrant publication in a widely visible interdisciplinary journal.

2. This title is not only misleading in intent but also in fact, some 'self-evident' others more nuanced. It is not 'creep' that is necessarily being measured but 'speed' or 'velocity' and these words are used in the text. Creep is associated here with rock glaciers, as in text and as stated but also with glacier ice deformation. Glaciers can slide to give a surface velocity.

We now modified the title and avoid the term 'creep' throughout the text, unless where citing other studies.

3. The linkage of rock glacier to 'permafrost creep' should be abandoned. This

has long been used as the only mechanism for rock glacier flow, usually following citation of the book Rock Glaciers by D. Barsch (here reference number 14). Barsch denies the existence of the 'glacier ice cored rock glacier'. In particular the work of Potter at Galena Creek RG where glacier ice has been shown to be the core of the rock glacier at Galena Creek and at Sulphur Creek RG the recent work by Meng et al (cited in the paper as ref 27) not only gives velocity data (was this compared to that obtained in this paper?). In other words, the authors use rock glacier always as being indicative of permafrost rock glaciers but omit references to evidence from 'disproving cases'. This has been a situation for far too long in the literature. Here the glacier ice formation is blatantly denied but still uses RG that do have glacier cores in the 'analysis'.

To our best understanding the term 'permafrost' does not exclude bodies of frozen debris with a core of glacier ice and the term is, in contrast to the term 'glacier', mainly a thermal term. Still, to avoid confusion we tried now to rephrase our text throughout and to direct the focus on the changes in speed, which we consider our most important results. See also our general response at the top of this response letter.

We indeed compared our measurements to the great and openly available ones by Meng et al. (2023), see last subsection of the Method section ('Comparison to other velocity measurements'). We also make our detailed speed data now accessible in a data repository along with other information in the Supplement enabling others to reproduce our results and to perform own detailed comparisons.

4. Any body with ice (glacier or cryogenic) can 'flow' or 'creep' but the creep rate (velocity) depends upon more than 'temperature'. Thickness of the ice mass and surface slope provide the main stress fields. So, a glacier might give the same results as supposed for a 'permafrost rock glacier'. But the problem really is that the data analyses are far too simplistic, other site specific conditions should be analysed. Further, if the rock glaciers are buried glaciers then sliding, rather than flowing, might be the main driver of the differences.

We formulated now more neutral throughout the text, well acknowledging that we do not know the internal structure and internal kinematics of the rock glaciers investigated.

Our comment list did not contain a comment #5 and we assume #6 should read #5, etc.

6. But what are these differences? Why does Cardinal Mtn N have the largest surface speed change compared with Cardinal Mtn S have one of the lowest? This alone might suggest that there is more in the variability than 'temperature'. Indeed, several meteorological parameters should be used and investigated. This leads to

We extended now the respective parts of the discussion, considering the different factors that are thought to influence rock glacier speed. We add substantially more meteorological data in the Supplement (as also suggested by referee #2). We also add analysis, discussion and figures on the relation between met data and our rock glacier speed.

7. The data analysis is overly simplistic. In fact, the paper does not comply with any aspects of FAIR data principles. Ok, have a graph but what are the data from which the results are plotted? Use of median might be more 'robust' but there are other descriptive statistics that could be used for the analysis.

We make now available the individual point speeds for all rock glaciers and the data of the graph Fig. 2. For analyzing the rock glacier speeds over time and their changes we use period medians (red lines), period means (middle of light blue areas, as now clarified), notches (which are derived from interquartile ranges and thus involve percentiles), an estimate of the error of speed changes between periods, and a statistical significance test of the latter. We have experimented with other statistical descriptors (e.g., percentiles, whiskers) but believe the current set of statistical descriptors covers the most important statistical aspects behind our time series while maintaining readability of Fig. 2. In addition, the stable ground (=co-registration) errors are given.

8. Also with respect to FAIR, the locations of rock glaciers are given only by toponyms. A proper geolocation is required. Using a decimal latitude-longitude [dLL] identifier for rock glacier locations allows the reader to locate the feature in, for example Google Earth. This might help in interpretation, what might the differences be between Cardinal Mountain N/S. From the peak at [36.9996,-118.4148], there are several contenders for 'north' but I cannot see what is meant by 'south'. This is not presenting data that are findable, accessible, interoperable or re-usable. This is required in any transparent data analysis. No attempt to be transparent is made in this paper.

We now substantially expanded the Supplement, among others with one separate sheet for each individual rock glacier, including airphoto, measurement locations, descriptive data and we provide the speed time series data.

But fundamentally, it is the continued scientific lack exactitude of the data presented. Support for the contention suggested by the title is not upheld.

We hope the modifications made to the manuscript and motivated in our above responses improved the paper. See also general response at the top of this response letter.

End of response

REVIEWERS' COMMENTS

Reviewer #2 (Remarks to the Author):

The authors have during the revision taken into account the comments from the reviewers, they have largely increased readability, transparency and reproducibility. They have given more details regarding data analysis and the interpretation.

This work is a significant contribution to the field, and similar fields.

The revised manuscript is in my opinion ready for publication.

Best regards,

Harald Øverli Eriksen

Reviewer #3 (Remarks to the Author):

Thank you for doing the changes and modifications.

Hi however, I would ask that you include [dLL] geolocations for the RG used in the analysis in the paper (not just in the Supplementary). Ideally each place name label should have its [dLL] below the toponym. However, I see no reason why they should not be given as a list in the caption for Figure 1. This would make the locations both easily identifiable and, importantly, digitally readable in the MS text.

for example Galena Creek: [44.6502,-109.7907]